

# SEF-UNet: advancing abdominal multi-organ segmentation with SEFormer and depthwise cascaded upsampling

Yaping Zhao[1], Yizhang Jiang[1], Lijun Huang[2] and Kaijian Xia[3,4]

[1] School of Artificial Intelligence and Computer Science, Jiangnan University, Wuxi, Jiangsu, China
[2] Department of Medical Imaging, The Changshu Affiliated Hospital of Soochow University, Suzhou, Jiangsu, China
[3] Department of Scientific Research, The Changshu Affiliated Hospital of Soochow University, Suzhou, Jiangsu, China
[4] Changshu Key Laboratory of Medical Artificial Intelligence and Big Data, Suzhou, Jiangsu, China

Corresponding author
Kaijian Xia, kjxia@suda.edu.cn

## ABSTRACT

The abdomen houses multiple vital organs, which are associated with various diseases posing significant risks to human health. Early detection of abdominal organ conditions allows for timely intervention and treatment, preventing deterioration of patients' health. Segmenting abdominal organs aids physicians in more accurately diagnosing organ lesions. However, the anatomical structures of abdominal organs are relatively complex, with organs overlapping each other, sharing similar features, thereby presenting challenges for segmentation tasks. In real medical scenarios, models must demonstrate real-time and low-latency features, necessitating an improvement in segmentation accuracy while minimizing the number of parameters. Researchers have developed various methods for abdominal organ segmentation, ranging from convolutional neural networks (CNNs) to Transformers. However, these methods often encounter difficulties in accurately identifying organ segmentation boundaries. MetaFormer abstracts the framework of Transformers, excluding the multi-head Self-Attention, offering a new perspective for solving computer vision problems and overcoming the limitations of Vision Transformers and CNN backbone networks. To further enhance segmentation effectiveness, we propose a U-shaped network, integrating SEFormer and depthwise cascaded upsampling (dCUP) as the encoder and decoder, respectively, into the UNet structure, named SEF-UNet. SEFormer combines Squeeze-and-Excitation modules with depthwise separable convolutions, instantiating the MetaFormer framework, enhancing the capture of local details and texture information, thereby improving edge segmentation accuracy. dCUP further integrates shallow and deep information layers during the upsampling process. Our model significantly improves segmentation accuracy while reducing the parameter count and exhibits superior performance in segmenting organ edges that overlap each other, thereby offering potential deployment in real medical scenarios.

# INTRODUCTION

The abdomen, as a critical region of the human body, encompasses several vital organs, including the stomach, liver, gallbladder, pancreas, spleen, and kidneys. The health of these organs directly impacts overall physiological functions and is closely associated with various common diseases such as gastric ulcers, liver cirrhosis, gallbladder inflammation, and pancreatitis, among others (*Sykes, 2014*). Abdominal-related diseases have a relatively high incidence, and many patients may experience asymptomatic abdominal conditions. Early detection of the relevant conditions of abdominal organs would contribute to timely intervention and treatment, preventing the progression of the disease and avoiding adverse consequences (*Senkyire & Liu, 2021*; *Shojaee, Sabzghabaei & Heidari, 2020*).

In the field of medicine, various imaging techniques are widely used for the detection and diagnosis of abdominal diseases. Among them, computed tomography (CT) stands out as an advanced medical imaging technology that utilizes X-rays and computer technology to obtain detailed three-dimensional cross-sectional images of the human body. CT technology can provide structural information for multiple organs such as the liver, kidneys, spleen, and pancreas. By segmenting CT images, the anatomical structures of various abdominal organs can be accurately depicted. This plays a crucial role in surgical planning, clinical decision-making, radiation therapy, and computer-aided diagnosis, among other applications (*Ma et al., 2021*; *Wang et al., 2019*).

Accurate segmentation and localization of different organs in abdominal images assist doctors in diagnosing lesions and diseases of abdominal organs more precisely. Before surgical procedures, doctors need a thorough understanding of the patient's anatomical structure to plan the surgery effectively. High-resolution three-dimensional images generated through abdominal organ segmentation allow doctors to comprehensively grasp the patient's abdominal anatomy, enabling more precise surgical planning. For patients with chronic diseases or those requiring regular treatment, abdominal multi-organ segmentation techniques can monitor disease progression and treatment effects. Doctors can assess the effectiveness of treatment by comparing images at different time points and adjust treatment plans accordingly. In organ transplant surgeries, abdominal multi-organ segmentation helps doctors evaluate the compatibility of organs between the patient and the donor, ensuring minimal organ damage during the surgical process. Overall, abdominal multi-organ segmentation technology plays a crucial role in improving the accuracy of medical imaging diagnosis, guiding treatment and surgical planning, contributing to enhanced patient outcomes.

However, the anatomical structure of abdominal organs is relatively complex, with mutual occlusion and similar features among organs, making the segmentation task more challenging (*Selver, 2014*). Firstly, due to the typically low contrast of soft tissues, boundaries between organs such as the liver, pancreas, and stomach are often indistinct, leading to potential errors. Secondly, there is significant variation in the shapes and positions of abdominal organs among different patients, and disparities in imaging scanners and CT phases can result in noticeable differences in organ appearances. Consequently, the generalization capability of universal models across different individuals is challenged.

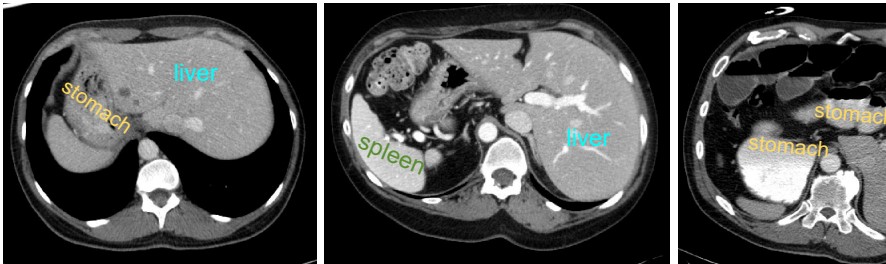

**Figure 1** Abdominal CT slice organ examples: (A) indistinct boundary between the liver and stomach; (B) stomach with multiple disconnected segments, possibly leading to segmentation omission; (C) enhanced contrast of vascular tissue within the liver parenchyma, potentially causing excessive segmentation.

Figure 1 illustrates some challenging examples, indicating that personalized and adaptive segmentation methods may be required to account for individual variations.

The development of abdominal multi-organ segmentation has undergone several stages. Early methods were primarily based on traditional image segmentation techniques, including threshold segmentation, edge detection, and mathematical morphology. Although these methods may perform well in some simple contexts, their accuracy and robustness are limited in complex medical images.

With the advancement of computer vision and machine learning, researchers began utilizing image features for segmentation. These methods leverage image features such as texture, shape, intensity, *etc.*, combined with machine learning algorithms, including support vector machine (SVM) (*Mountrakis, Im & Ogole, 2011*) and random forests, for abdominal multi-organ segmentation. These approaches have to some extent improved the accuracy of segmentation.

Entering the 21st century, the rise of deep learning has had a revolutionary impact on image segmentation. Deep learning models, especially convolutional neural networks (CNNs), possess the capability to learn high-level features in images, enabling more accurate medical image segmentation. The adoption of deep learning methods, particularly network architectures like UNet (*Ronneberger, Fischer & Brox, 2015*), SegNet (*Badrinarayanan, Kendall & Cipolla, 2017*), and others, has significantly improved the accuracy and efficiency of medical image segmentation.

Transformer (*Vaswani et al., 2017*) is a groundbreaking innovation in the field of deep learning, with its core idea being the Multi-Head Attention mechanism. By establishing global dependencies in sequential data, this model captures contextual information more effectively and has achieved significant success in natural language processing. Vision Transformer (*Dosovitskiy et al., 2020*) takes a novel approach by treating images as one-dimensional sequences, analogous to words, and processes them using a transformer model. This has opened a new era for the application of transformer models in the field of computer vision.

When applying Transformer to the field of medical image segmentation, directly using the hidden features obtained from the Transformer encoder for upsampling to

full resolution does not yield satisfactory results. Therefore, drawing inspiration from the UNet architecture, an effective strategy involves utilizing convolutional layers and upsampling in the decoder part to restore low-resolution feature maps to the original resolution. Simultaneously, incorporating the UNet concept allows for the fusion of high-level semantic information from the encoder, thus better capturing both global and local information in the image.

In state-of-the-art deep learning models, researchers have conducted studies on combining Transformer with the most classic medical image segmentation model, UNet, exploring how to integrate them for medical image segmentation tasks (*Xiao et al., 2023*; *Aljabri & AlGhamdi, 2022*). TransUNet, proposed by *Chen et al. (2021)* is the first work that incorporates Transformer blocks into the UNet architecture. They employ a hybrid CNN-Transformer architecture to extract features, then use a cascaded upsampler (CUP) to decode hidden features, combining them with different high-resolution CNN features from the encoder to achieve accurate segmentation of medical images.

However, the TransUNet model faces certain challenges in terms of computational resources. Its substantial number of parameters and computational requirements may result in limitations in memory and computing, especially when dealing with large-scale datasets or extensive models. Training a high-performing TransUNet model demands a significant amount of training data, training resources, and time.

In recent research, scholars have explored replacing the attention modules in transformers with alternative modules. For instance, *Tolstikhin et al. (2021)* utilized spatial MLPs as token mixers, and *Lee-Thorp et al. (2021)* employed Fourier transforms as a replacement for attention. These alternative methods have demonstrated satisfactory performance in image tasks. Building on these studies, *Yu et al. (2022)* synthesized experiences from similar models, suggesting that the crucial element in the Transformer is not the attention-based token mixer but a general architecture beyond the token mixer. They proposed a non-parametric operator, pooling, as a token mixer and instantiated a specific model called PoolFormer. PoolFormer outperforms traditional Transformers and some MLP-based models, such as ResMLP (*Touvron et al., 2022*) and DeiT (*Touvron et al., 2021*), with a substantial reduction in the number of parameters.

Inspired by these studies, we propose a SEF-UNet model. This model integrates the principles of MetaFormer, utilizing squeeze-and-excitation modules and depthwise separable convolution to instantiate a new MetaFormer model, named SEFormer, serving as the encoder for feature extraction. Simultaneously, we introduce deep convolutional layers into the decoder, presenting a depthwise cascaded upsampling module. The purpose of this is to strengthen the capture of local details and texture information while merging features from the encoder, thereby enhancing segmentation accuracy. Through experiments, our model has demonstrated superior performance on a multi-organ abdominal segmentation dataset compared to the aforementioned models, with a reduction in the model parameter count.

The main contributions of this article are as follows:
- We introduced the SEFormer structure, abstracting the architecture in Transformer beyond the multi-head self-attention module. We employed SE depthwise separable

convolution as a token mixer, recalibrating features to enhance the weight of relevant information, thereby improving the model's performance.

- We incorporated depthwise convolution into the decoder structure, creating the depthwise cascaded upsampling module (dCUP). By stacking multiple dCUPs, we achieved multi-level upsampling of deep features, further enhancing feature fusion to capture local details and texture information, thus improving segmentation effectiveness.
- In contrast to many models utilizing Transformer structures, our proposed model significantly reduces the number of parameters. This reduction provides the potential for deploying the model in practical medical applications.
- Through experiments, our model demonstrated a noticeable improvement in segmentation accuracy, particularly excelling in the segmentation of organ edges that are mutually occluded. This further substantiates the effectiveness of the model.

This article exhibits a clear structure. In the first section, we review the challenges in abdominal organ segmentation, providing a detailed account of the motivation for the study, the evolution of application methods, and the main contributions of the article. The second section reviews two highly relevant architectures to this study, SE-Net and MetaFormer, offering readers necessary background knowledge. The third section elaborates on the architecture of the proposed SEF-UNet model, providing detailed explanations for the two key components, SEFormer and dCUP. The fourth section outlines the experimental process, encompassing experimental design, results presentation, and data analysis, offering readers a comprehensive understanding of the model's performance. Finally, the fifth section provides a summary and conclusion, emphasizing the significance of the research and the achieved results.

## RELATED WORK

### Squeeze-and-excitation networks

Squeeze-and-Excitation (SE) block (*Hu, Shen & Sun, 2018*) is an architectural unit that focuses on channel relationships. This module selectively enhances the weight of informative features and suppresses less useful features by learning global information.

The SE block consists of two main stages: Squeeze and Excitation. For an input $X$, a transformation $\mathbf{F}_{tr} : \mathbf{X} \rightarrow \mathbf{U}$ is performed, where $X \in R^{H' \times W' \times C'}$ and $U \in R^{H \times W \times C}$. This transformation can be operations such as convolution or convolution sets. The SE module is introduced into this transformation process to recalibrate the features.

Firstly, through the squeeze operation, features are aggregated into a $H \times W$ feature map, consolidating global information to generate channel descriptors. Following is the excitation operation, where the model activates specific channels by learning sample-specific activations through a channel-dependent gating mechanism. Subsequently, the obtained activation data is used to weight the feature map $U$. It is noteworthy that, since the size and number of channels of the feature map remain unchanged, the output can be directly passed to subsequent layers without significant adjustments to the original model's structure. The introduction of the SE module contributes to improved model performance, allowing for more effective capture and utilization of global contextual information.

The entire process can be represented by the following formula:

$$\text{SE}(x) = \sigma\left(W_2\delta(W_1\text{AvgPool}(x))\right) \cdot x \tag{1}$$

For input $x$, the average value of each channel is obtained through global average pooling to acquire global information for each channel. This helps reduce information within each channel, incorporating global contextual information into the description of each channel. The aforementioned part constitutes the Squeeze operation. The Excitation stage includes the first fully connected layer, ReLU activation function, the second fully connected layer, and the Sigmoid function. The first fully connected layer is a compressed fully connected layer, reducing the number of channels from $C$ to $\frac{C}{r}$. The second fully connected layer is the excitatory fully connected layer, restoring the number of channels from $\frac{C}{r}$ to $C$. The goal of these two layers is to learn the weights for each channel, enhancing or diminishing the feature response of that channel.

By introducing the Squeeze and Excitation stages, SENet can adaptively learn the importance of each channel and capture key information more effectively through updated feature maps. This has led to significant performance improvements for SENet in various image tasks.

The SE block is computationally lightweight, providing noticeable improvements to the model with minimal additional computation. The introduction of the SE module helps the network capture crucial information more efficiently from input features, enhancing model performance. This module is typically integrated into various layers of CNNs to enhance the network's representational capabilities.

## MetaFormer

MetaFormer (*Yu et al., 2022*) is a versatile architecture that abstracts out parts of the Transformer except for the multi-head self-attention module. A MetaFormer block consists of two residual blocks, where the first residual block can be represented as:

$$X' = \text{TokenMixer}(\text{Norm}(X)) + X \tag{2}$$

For input $X$, it first passes through the $\text{Norm}(\cdot)$ layer, representing the normalization layer. Following that is $\text{TokenMixer}(\cdot)$, used for token information mixing, with various implementations such as Identity Mapping, Random Mixing, Separable Convolution, Attention, *etc.*

The second residual block primarily consists of a two-layer MLP and a non-linear activation layer, expressed as:

$$Y = W_2(\sigma(W_1(\text{Norm}(X')))) + X' \tag{3}$$

Here, $W_1 \in R^{C \times rC}$ and $W_2 \in R^{rC \times C}$ are the learnable parameters for the two fully connected layers, and $\sigma(\cdot)$ is the non-linear activation function, typically using ReLU or its improved versions.

## METHOD

The current methods for abdominal multi-organ segmentation have made some progress, but there are still challenges and limitations:

1. The anatomical structure of abdominal organs is relatively complex, with issues such as mutual occlusion and unclear boundaries between organs. Existing methods have room for improvement in the accuracy of organ segmentation edges.

2. In practical medical applications, especially in real-time scenarios like surgical navigation, models must possess real-time capabilities and low latency. This requires abdominal organ segmentation models to achieve real-time performance and low latency while maintaining high accuracy. Striking a balance between reducing the number of parameters and improving segmentation accuracy is a challenge. Larger models may offer better performance, but they could also lead to increased computational and memory requirements, making them less suitable for real-time applications. Therefore, it is crucial to find a balance between the number of parameters and performance.

Our research focuses on enhancing the accuracy of abdominal organ edge segmentation, optimizing the model structure to meet real-time and low-latency requirements, and finding a better balance between the number of parameters and segmentation accuracy.

We have designed and proposed a novel model with a basic structure in the form of UNet, consisting of encoder and decoder with four-stage architectures each. In the encoder, the ith block consists of a downsampling module and Li SEFormer blocks. The entire four-stage architecture is responsible for gradually extracting features at different levels. The first stage focuses on extracting low-level features, typically related to the original information of the input data, such as edges and textures in images. The second stage builds upon the first stage to further extract mid-level features, including more complex structures like object shapes. The third and fourth stages can extract higher-level abstract features, such as object categories and scene semantics.

This multi-stage architecture allows the network to progressively learn and integrate features at different levels, enhancing the model's understanding of the images. The feature maps obtained from the first three stages are transmitted to the decoder through skip connections, while the output from the fourth stage serves as input to the decoder. After passing through four dCUP modules, the decoder generates an output of the same size as the original image. Throughout this process, features from the encoder are fused with the feature information in the decoder's upsampling, effectively utilizing both shallow and deep features to preserve more detailed information and improve the model's understanding of image content. We named this model SEF-UNet, and its complete structure is illustrated in Fig. 2.

The main innovations of this model lie in the SEFormer block and dCUP block. The SEFormer block combines depthwise separable convolution with the SE structure. Compared to regular convolution, depthwise separable convolution significantly reduces the number of parameters and computations. The SE structure weights the feature mappings at different levels within the network, enabling the network to recalibrate the feature response of each channel. This enhances useful features and suppresses unimportant ones, more effectively capturing crucial information in the input data. This is particularly useful in situations where organs obscure each other, as the model needs to distinguish which features belong to the foreground organ and which belong to the obscured parts, thereby improving

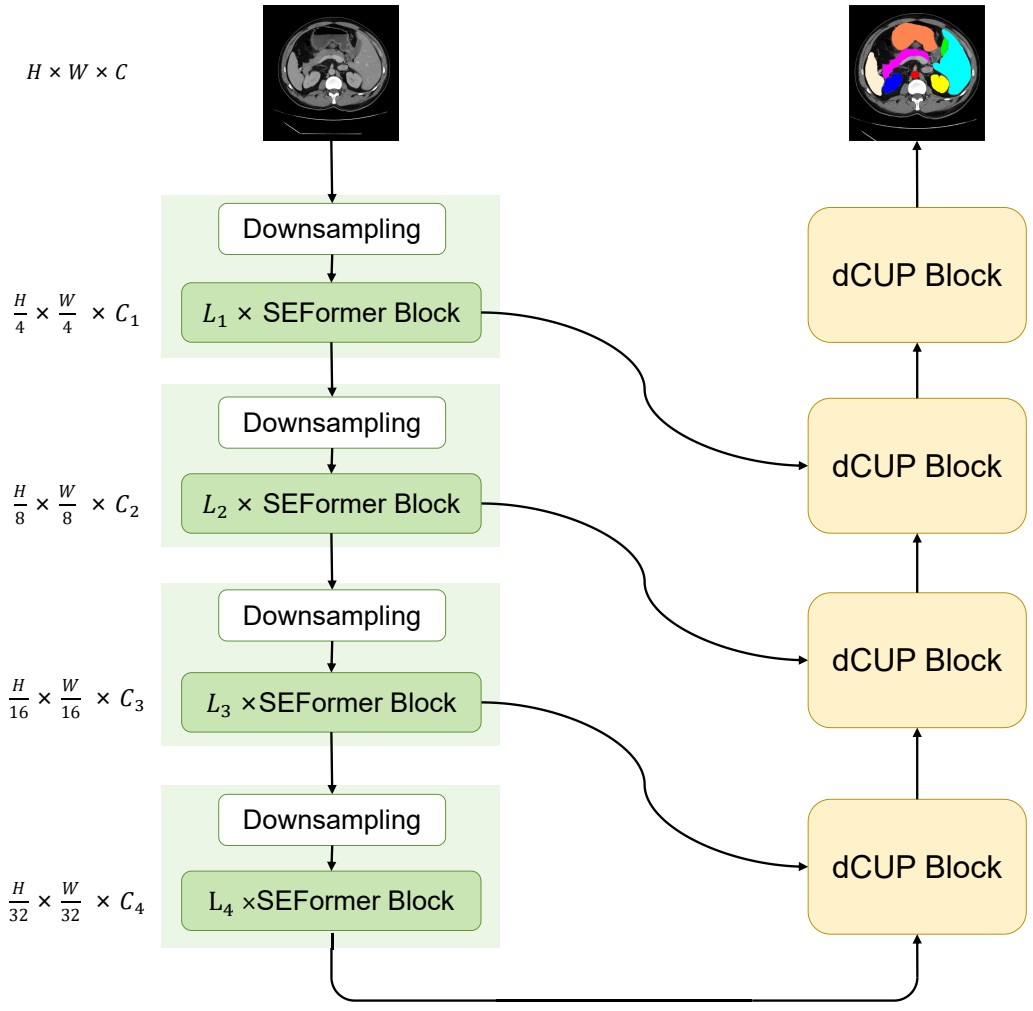

**Figure 2   The overall architecture of SEF-UNet.**

the accuracy of organ segmentation edges. The combination of these two structures not only enhances the model's information processing capabilities but also significantly reduces the number of parameters compared to many other models that heavily use Transformers. The dCUP block, built upon the upsampling structure, incorporates depthwise convolution. With a minimal increase in the number of parameters and computations, it greatly enhances the ability to restore details and semantic information from the original input image.

## SEFormer block

The overall structure of the SEFormer block is illustrated in Fig. 3. We use the MetaFormer (*Yu et al., 2022*; *Yu et al., 2023*) architecture as the backbone and instantiate the token mixer with a combination of the Squeeze-and-Excitation module and depthwise separable convolution. The use of depthwise separable convolution retains good performance while significantly reducing the number of parameters and computations. Simultaneously, the

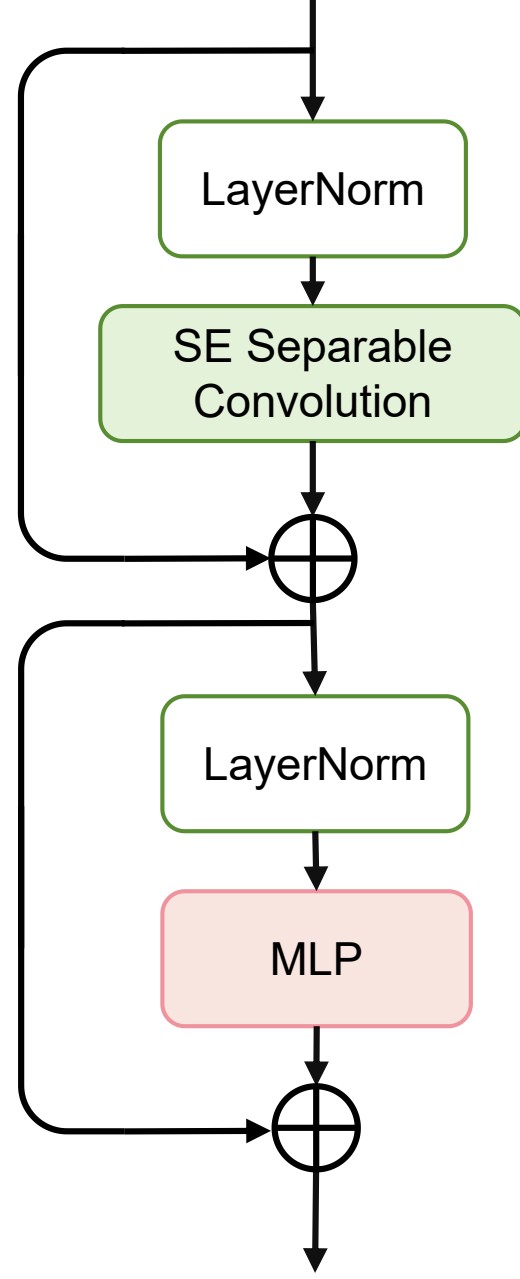

**Figure 3** **Schematic of SEFormer.**

SE module captures crucial information in the input features, enhancing the network's expressive capabilities.

Figure 3 depicts the complete structure of the SEFormer block. The expression for the SEFormer block is as follows:

$$X' = SESepConv(Norm(X)) + X \qquad (4)$$

$$SEFormerBlcok(X) = X' + MLP(Norm(X'))   \tag{5}$$

This module consists of two residual structures. First, the input $X$ is normalized using LayerNorm ($Norm(\cdot)$). Subsequently, after passing through the $SESepConv(\cdot)$ module, the output is added to the original input to obtain an intermediate value $X'$. This intermediate value is then normalized again through LayerNorm and processed through a Multilayer Perceptron (MLP) containing two fully connected layers. Finally, the output of the MLP is added to the unprocessed intermediate value $X'$ to obtain the final output of the module.

The SE Separable Convolution is the core unit of this module, as shown in the structure in Fig. 4. The entire structure mainly consists of pointwise convolution, depthwise convolution, and the branched Squeeze-and-Excitation structure.

The specific expression for this module is as follows:

$$SEConv(X) = Conv_{pw2}(SE(Conv_{dw}(\sigma(Conv_{pw1}(X)))))   \tag{6}$$

where $Conv_{dw}(\cdot)$ represents depthwise convolution, independently convolving each channel of the input. $Conv_{pw1}$ and $Conv_{pw2}$ denote pointwise convolution, merging the output of depthwise convolution through a $1 \times 1$ convolution.

For the input $X$, the calculation formula to obtain the output $Y$ through depthwise convolution is as follows:

$$Y_{i,j,c} = \sum_{m,n} X_{i \times s+m, j \times s+n, c} \times K_{m,n,c}   \tag{7}$$

where $i, j$ are the spatial coordinates of the output tensor, $c$ is the channel index, $s$ is the stride of depthwise convolution, $m, n$ are the coordinates of the convolution kernel for depthwise convolution, $X_{i,j,c}$ is the element of the input tensor, and $K_{m,n,c}$ is the parameter of the depthwise convolution kernel.

The calculation formula for pointwise convolution is:

$$Y_{i,j,c} = \sum_{k} X_{i,j,k} \times K_{1,1,k,c}   \tag{8}$$

where $K_{1,1,k,c}$ is the convolutional kernel parameter for pointwise convolution, and the other parameters are consistent with Eq. (7).

The Squeeze-and-Excitation structure mainly consists of two stages: Squeeze and Excitation. Assuming the input feature map $X$ has dimensions $H \times W \times C$, where $H$ and $W$ represent the height and width of the input feature map respectively, and $C$ represents the number of channels. During the squeezing stage, global average pooling is used to capture the global information of each channel of the input feature map $X$.

$$z_c = \frac{1}{H \times W} \sum_{i=1}^{H} \sum_{j=1}^{W} x_{cij}   \tag{9}$$

Here, $z_c$ is the global average of the $c$-th channel, and $x_{cij}$ is the value of input $X$ at the $c$-th channel, where the row and column indices are $i$ and $j$, respectively.

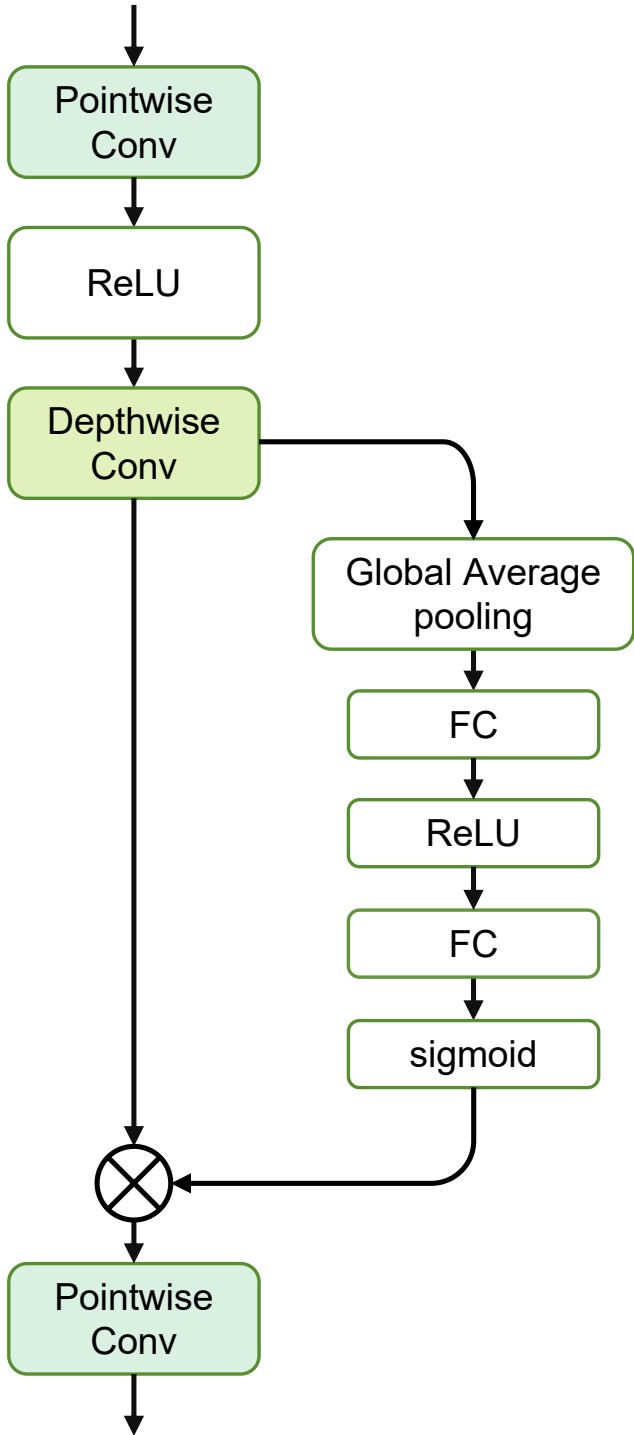

**Figure 4** Structure of the SE separable convolution.

In the Excitation stage, two fully connected layers are introduced, and the Sigmoid activation function is used to learn the weights of each channel. These two fully connected layers are referred to as Squeeze and Excitation, respectively. The first fully connected layer is used to compress the number of channels, reducing the global average value $z_c$ of each channel by a compression ratio $r$:

$$s_c = \sigma(W_1 \cdot z_c) \tag{10}$$

where $W_1$ is the weight matrix of the first fully connected layer, $\sigma$ is the activation function, typically chosen as ReLU.

The second fully connected layer is used to restore the number of channels by exciting the original feature map with the learned weights $s_c$:

$$f_c = \sigma(W_2 \cdot s_c) \tag{11}$$

where $W_2$ is the weight matrix of the second fully connected layer, and $\sigma$ is the Sigmoid activation function.

Finally, by applying the weights of each channel to the original feature map, the updated feature map is obtained:

$$y_{cij} = f_c \cdot x_{cij} \tag{12}$$

where $y_{cij}$ represents the value at position $(c, i, j)$ in the updated feature map.

Therefore, the overall representation of the Squeeze and Excitation process can be expressed as:

$$SE(X) = \sigma(W_2\sigma(W_1(AvgPool(X)))) * X \tag{13}$$

## dCUP as decoder

To further enhance model computational efficiency, reduce parameter count, and improve segmentation accuracy, we designed a depthwise cascaded upsampling (dCUP) structure, which utilizes a depthwise convolution layer. This structure is constructed using components shown in Fig. 5.

The dCUP architecture comprises upsampling layers, feature fusion layers, convolution layers, activation functions, *etc.* The upsampling layers aim to increase the resolution of the feature map, gradually improving resolution by stacking multiple dCUP blocks to generate an output of the same size as the input image. Feature fusion involves concatenating feature maps from the corresponding stages of the encoder with the current feature map of the decoder stage. The concatenated data undergoes a convolution layer to adjust the channel count to the target channel count, followed by batch normalization and ReLU operations to enhance network training stability. Finally, through a depthwise convolution layer, features are further fused to capture local details and texture information. The expression for the entire process is as follows:

$$Y = Conv_{dw}(\sigma(Norm(Conv(cat(feature, Up(X)))))) \tag{14}$$

where $Up(\cdot)$ represents bilinear interpolation upsampling, which is concatenated with features from the encoder using the cat method. $\sigma(\cdot)$ is the activation function, and

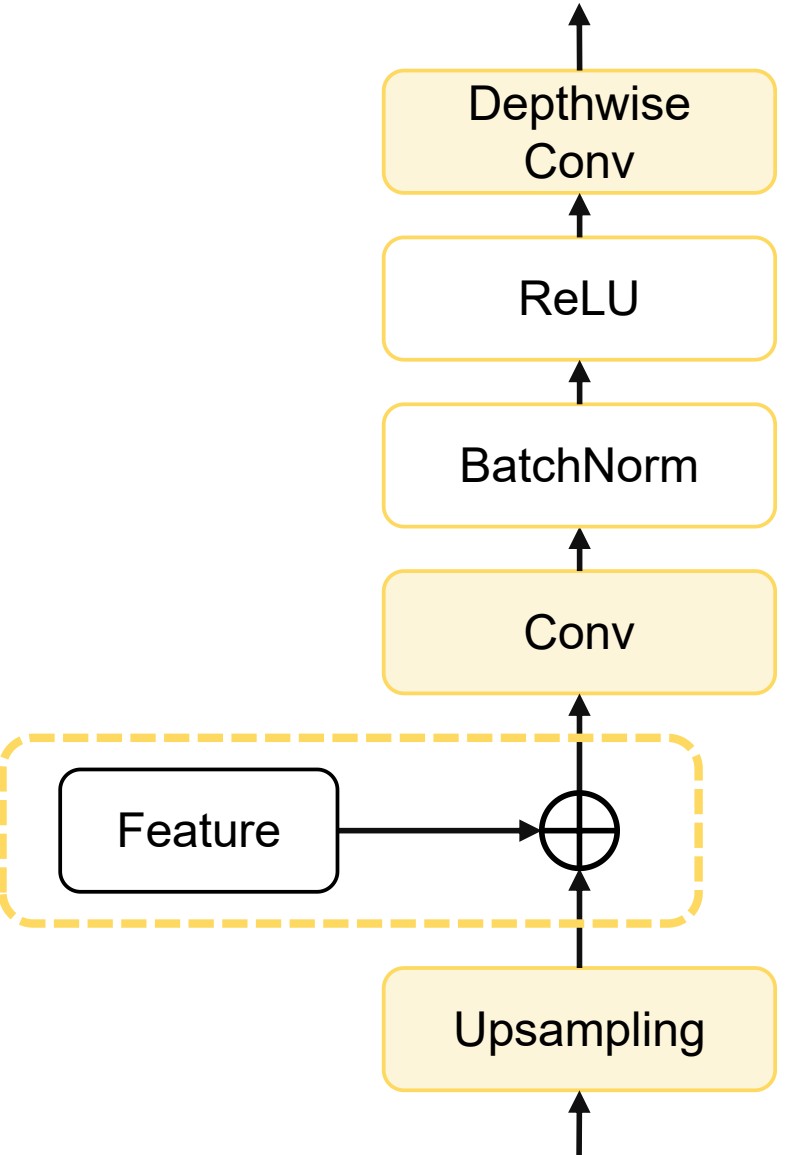

**Figure 5** Architecture of dCUP.

$Norm(\cdot)$ is the batch normalization operation. Since there are three sets of features from the encoder, only the first three dCUPs need to receive features and perform feature fusion operations, while subsequent dCUPs do not need to combine with features but proceed directly to the subsequent steps.

The decoder structure of the entire model consists of four dCUPs. Such upsampling blocks can simultaneously achieve upsampling, feature concatenation, and feature processing. The introduction of the depthwise convolutional layer significantly enhances the ability to restore details and semantic information of the original input image with only a small increase in the number of parameters and computational complexity.

The entire model extracts high-level semantic information from the input image through the encoder, retains low-level detail information through skip connection, and gradually combines these pieces of information in the decoder to generate the final segmentation result.

# EXPERIMENTS

## Dataset

Our dataset is Multi-Atlas Labeling Beyond the Cranial Vault (BCV) (https://www.synapse.org/#!Synapse:syn3193805/wiki/217789). The dataset has a total of 30 abdomen CT scans taken during portal venous contrast phase. The pixel size of the scans is $512 \times 512$ and the slices ranges from 85 to 198. Field of view size is approximately $280 \times 280 \times 280$ mm$^3-500 \times 500 \times 650$ mm$^3$, and the voxel spatial resolution of ($[0.54\sim0.54] \times [0.98\sim0.98] \times [2.5\sim5.0]$) mm$^3$

The BCV dataset consists of a total of 13 categories, and we have selected 8 abdominal organs for the segmentation task. These 8 organs are the spleen (Sp), right kidney (RK), left kidney (LK), gallbladder (Ga), liver (Li), stomach (St), aorta (Ao), and pancreas (Pa). We selected these eight categories because these organs are of significant importance in clinical diagnosis and treatment. Additionally, this selection aligns with the multi-organ segmentation tasks in existing literature, facilitating a fair and effective comparison with existing methods.

In all 30 samples, complete annotations for the 8 classes are provided. We randomly chose 21 samples for training, and the remaining 9 samples were used for testing. The 21 training samples were divided into 2,698 slices horizontally along the axial direction. All experimental results were averaged over the 9 test samples.

Inspired by *Xie et al. (2021)*, for the original CT dataset, we first truncated the Hounsfield Unit (HU) values to the range $[-125, 275]$ and then normalized them to scale the values between $[0, 1]$. During the training process, we applied random flips, rotations, and other operations to the input slice data to increase the diversity of the training set. Subsequently, the image size was resized to $224 \times 224$ to obtain fixed-size inputs for further training.

## Implementation details

The SEF-UNet model is implemented in Python 3.7 with PyTorch 1.13. It utilizes the SGD optimizer with a learning rate of 0.01, momentum set to 0.9, and weight decay of $1e-4$. The batch size for all experiments is set to 24, and the input resolution is $224 \times 224$. Each model is trained for 150 epochs on 2,698 slices, with each epoch containing 113 iterations, resulting in a total of 16,950 iterations. The experiments were conducted on a single NVIDIA GeForce RTX 4060 GPU.

Other relevant parameters during training include: the encoder consists of four blocks, with the repetition count for each block, *i.e.*, $[L_1, L_2, L_3, L_4]$, set as $[3, 3, 9, 3]$, and the corresponding channel numbers for each block (downsampling module is used for adjusting image resolution and channel numbers, while the block is mainly used for feature extraction with unchanged channel numbers) set as $[64, 128, 320, 512]$.

The backbone network in the encoder is pretrained on ImageNet21k. The decoder stage uses four dCUPs with an upsampling factor of 2, and the output channel numbers for each upsampling block are [256, 128, 64, 16]. The first three dCUPs receive fused skip features from the encoder, while the last one does not receive fusion information and is mainly used for further feature processing.

The training data consists of processed 2D slices. During the prediction process, inference is performed slice by slice, and the predicted 2D slices are stacked together to construct the 3D prediction result.

The loss function used in the experiments is a combination of CrossEntropyLoss and DiceLoss. For a sample with N classes and true labels $y_1, y_2, \ldots, y_N$, and model output probability distribution $p_1, p_2, \ldots, p_N$, the loss function is defined as:

$$L_{seg} = (L_{CE} + L_{Dice})/2 = (-\sum_{i=1}^{N} y_i \cdot \log(p_i) + 1 - \frac{2\sum_{i=1}^{N} y_i \cdot p_i + \epsilon}{\sum_{i=1}^{N} y_i + \sum_{i=1}^{N} p_i + \epsilon})/2 \tag{15}$$

where $\epsilon$ is a small constant added for numerical stability.

CrossEntropyLoss effectively measures the matching degree between the model's output probability distribution and the true labels. On the other hand, DiceLoss provides a smoother computation, contributing to training stability. However, DiceLoss may appear overly optimistic in certain situations as it does not consider subtle differences between true labels and predictions. Therefore, we choose to combine these two losses to comprehensively consider both the matching degree of the model's output probability distribution with true labels and sensitivity to subtle differences, aiming for a more holistic model training approach.

## Result

We conducted experiments comparing our model with several traditional medical image segmentation models and advanced segmentation models. The models include the classic UNet (*Ronneberger, Fischer & Brox, 2015*), UNet++ (*Zhou et al., 2018*) that enhances UNet with more skip connections to improve multi-scale feature modeling, UNet3+ (*Huang et al., 2020*) that introduces attention mechanism and dense connections, as well as two important and classic models in the medical image segmentation field that incorporate Transformer: the hybrid CNN-Transformer model TransUNet (*Chen et al., 2021*) and the pure Transformer model SwinUNet (*Cao et al., 2022*).

Table 1 shows the average dice similarity coefficient (DSC), 95% average Hausdorff distance (HD) and segmentation results for each abdominal organ. The bar chart (Fig. 6) visually presents the average DSC for each organ. From the results, it can be observed that gallbladder and pancreas have relatively poorer segmentation results compared to other organs. This is because they overlap significantly with other organs, making it challenging to distinguish them from adjacent organs and other structures. The segmentation accuracy for other organs mostly exceeds 85%, averaging 89.56%, indicating good segmentation results.

The SEF-UNet model achieved the best results in terms of average DSC (83.90%) and average HD (15.64%). Compared to basic CNN models like UNet, UNet++, and

**Table 1** Comparison of results from multiple methods on the BCV dataset containing average dice score (%), average Hausdorff distance (mm) and average dice score for each organ (%).

| Method | DSC | HD95 | Organs | | | | | | | |
|---|---|---|---|---|---|---|---|---|---|---|
| | | | Ao | Ga | LK | RK | Li | Pa | Sp | St |
| UNet | 75.61 | 32.50 | 87.80 | 62.02 | 78.96 | 72.98 | 93.48 | 52.02 | 84.02 | 73.65 |
| UNet++ | 77.87 | 29.12 | **88.90** | 62.92 | 83.84 | 74.36 | 94.41 | 57.40 | 86.99 | 74.18 |
| UNet3+ | 77.16 | 28.67 | 87.37 | 65.62 | 84.98 | 82.76 | 92.59 | 55.60 | 82.42 | 65.97 |
| SwinUNet | 80.49 | 19.29 | 83.21 | **70.52** | 87.14 | 81.02 | 93.89 | 57.17 | 89.21 | 81.76 |
| TransUNet | 81.31 | 28.79 | 86.56 | 61.84 | 88.80 | 87.34 | **95.07** | 57.77 | **91.90** | 81.18 |
| Ours | **83.90** | **15.64** | 86.11 | 69.38 | **91.66** | **89.03** | 94.38 | **64.45** | 91.12 | **85.08** |

**Notes.**
The best results are shown in bold.

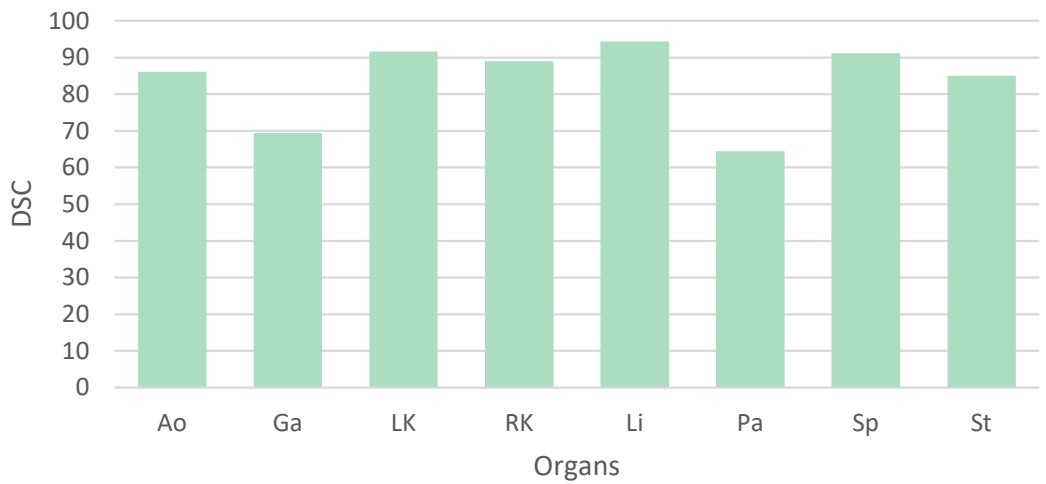

**Figure 6** Mean dice scores for image segmentation classes.

UNet3+, the SEF-UNet showed significant improvements in segmentation accuracy. It exhibited an 8.29% increase in average DSC compared to UNet and a 6.03% increase compared to the best-performing UNet++. Additionally, it demonstrated a reduction in Hausdorff distance by 13.03%–16.86%, surpassing a 40% decrease. When compared to the hybrid CNN-Transformer model TransUNet and the pure Transformer model SwinUNet, SEF-UNet not only showed improved average DSC by 2.59% and 3.41%, respectively, but also exhibited a reduction in HD by 13.15% and 3.65%, showcasing a clear advantage.

The SEF-UNet model has a parameter count of 24.65 M, placing it in the middle range when compared to other models. It has a comparable parameter count to the larger CNN model UNet3+, but it significantly reduces the parameter count by over 75% compared to the Transformer-based TransUNet. This optimization addresses the issue of excessive parameters in Transformer models, leading to improved segmentation performance without excessive computational resource consumption.

**Table 2   Comparison of GFLOPs, number of model parameters (M), and time taken per sample (minutes) between the experimental model and the comparative model.**

| Method | GFLOPs | Param | Test duration |
|---|---|---|---|
| UNet | 30.74 | 16.46 | 1.55 |
| UNet++ | 26.71 | **8.74** | 1.81 |
| UNet3+ | 154.03 | 25.74 | 1.92 |
| SwinUNet | 5.95 | 25.91 | 4.57 |
| TransUNet | 24.73 | 100.40 | 4.60 |
| Ours | **5.16** | 24.65 | **1.41** |

**Notes.**
The best results are shown in bold.

Additionally, to further validate the improvements of our experimental model in terms of real-time performance and low latency, we compared the GFLOPs and the average time taken to segment each test sample across all models, which is shown in Table 2. It can be observed that our model has the lowest GFLOPs, indicating the least number of floating-point operations required. Compared to the highest GFLOPs model, UNet3+, our model reduces the GFLOPs by 29.8 times.

Furthermore, we measured the average time taken by each model to process nine test samples from the BCV dataset. Our model takes only 1.41 min on average, making it the fastest model. This is more than three times faster than the slowest model, TransUNet. It is also important to note that a single test sample is 3D data, with its width and height resized to the target size during training, while the number of channels remains unchanged. The average number of channels for all samples is 125.9. The testing process involves processing a single 2D image and then combining them into a 3D image. Therefore, the average time taken by our experimental model to process a single 2D image is 0.67 s. These data can be found in Table 2.

In addition, we have presented the average DSC and HD of both the proposed experimental model and the comparative models for each test case (Table 3). The more intuitive comparative results are illustrated in Figs. 7 and 8. It can be observed that our experimental model exhibits the most consistent performance across all test cases, with both DSC and HD outperforming those of the other comparative models on the whole Fig. 9 displays the visual segmentation results of nine test cases. It can be observed that the segmentation boundaries are clear, indicating stable segmentation performance.

In summary, the experimental results indicate that, in medical image segmentation tasks, the SEF-UNet model exhibits significant advantages over CNN models, hybrid CNN-Transformer models, and pure Transformer models.

## Ablation experiment

To comprehensively validate the effectiveness of our experimental model, we conducted multiple sets of experiments. We explored different combinations of encoders, decoders, SEFormer, and dCUP within the SEF-UNet framework, and validated the model on the BCV dataset. Additionally, we investigated the impact of varying the number of stages in the model on segmentation performance. Furthermore, we used a larger resolution

**Table 3** Average dice score (%) and average Hausdorff distance (mm) for our method as well as the comparison model on each test sample.

| Sample | Evaluation indicator | Method | | | | | |
|---|---|---|---|---|---|---|---|
| | | Ours | UNet | UNet++ | UNet3+ | SwinUNet | TransUNet |
| 1 | DSC | **78.70** | 72.84 | 73.11 | 74.91 | 73.91 | 74.25 |
| | HD | 27.97 | 41.02 | 24.73 | **20.44** | 33.09 | 42.20 |
| 2 | DSC | **85.58** | 79.10 | 82.09 | 84.33 | 84.62 | 85.03 |
| | HD | 23.04 | 36.69 | 41.28 | 24.35 | **6.78** | 29.38 |
| 3 | DSC | **87.73** | 75.22 | 78.92 | 82.21 | 84.57 | 87.70 |
| | HD | **8.99** | 30.73 | 30.36 | 27.40 | 26.72 | 22.35 |
| 4 | DSC | **84.16** | 69.75 | 69.22 | 73.86 | 77.78 | 79.28 |
| | HD | **20.18** | 25.17 | 33.51 | 23.78 | 20.86 | 26.85 |
| 5 | DSC | **82.40** | 71.73 | 79.25 | 69.78 | 72.37 | 74.41 |
| | HD | **24.31** | 53.24 | 25.45 | 53.17 | 30.20 | 44.78 |
| 6 | DSC | **87.49** | 80.19 | 82.15 | 76.62 | 86.60 | 87.07 |
| | HD | **6.33** | 21.96 | 26.13 | 37.04 | 8.43 | 19.61 |
| 7 | DSC | **86.87** | 79.26 | 83.08 | 72.33 | 85.29 | 86.81 |
| | HD | **3.08** | 35.78 | 8.66 | 24.49 | 4.76 | 13.32 |
| 8 | DSC | **83.42** | 78.95 | 78.67 | 82.70 | 81.36 | 81.99 |
| | HD | **14.84** | 23.27 | 30.72 | 19.25 | 22.91 | 26.45 |
| 9 | DSC | **78.76** | 73.42 | 74.36 | 77.73 | 77.92 | 75.22 |
| | HD | **12.05** | 24.68 | 41.27 | 28.12 | 19.88 | 34.22 |

**Notes.**
The best results are shown in bold.

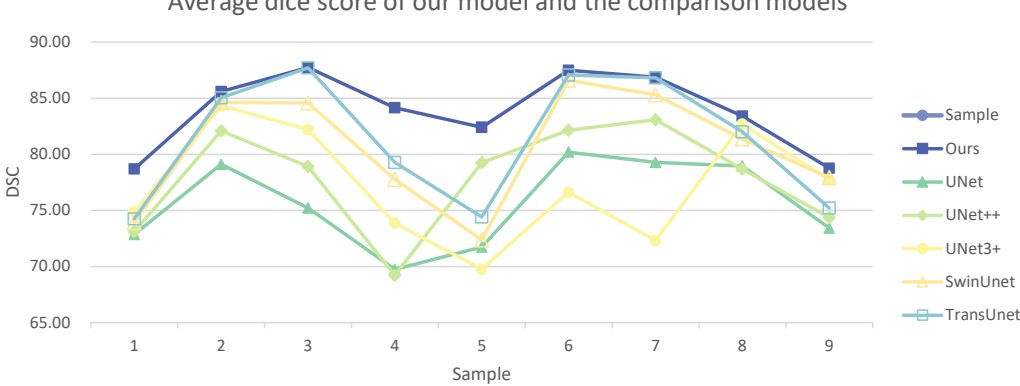

**Figure 7** Average dice score of our model and the comparison models.

of $384 \times 384$ as the model input to further explore the effectiveness of the SEF-UNet model. This series of experiments aims to thoroughly assess the model's performance under different configurations, ensuring its robustness and generalizability across various conditions.

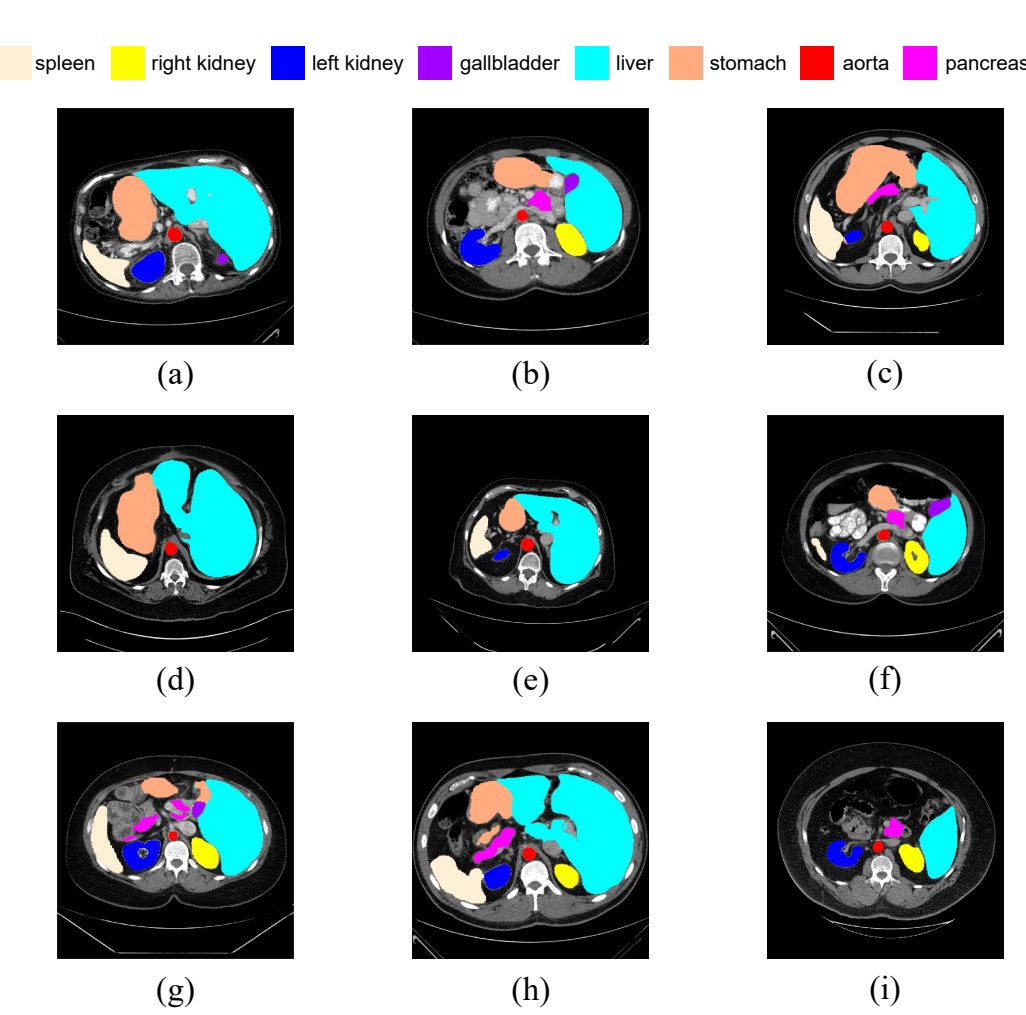

**Average hausdorff distance of our model and the comparison models**

Legend: Ours, UNet, UNet++, UNet3+, SwinUnet, TransUnet

X-axis: Sample (1–9), Y-axis: HD (0.00–60.00)

**Figure 8** Average Hausdorff distance of our model and the comparison models.

Legend: spleen, right kidney, left kidney, gallbladder, liver, stomach, aorta, pancreas

(a)     (b)     (c)

(d)     (e)     (f)

(g)     (h)     (i)

**Figure 9** (A–I) Visual segmentation results on nine test samples.

**Table 4** Comparative experiments exploring the validity of model encoder and decoder separately.

| Framework | | DSC | HD95 | Organs | | | | | | | |
|---|---|---|---|---|---|---|---|---|---|---|---|
| Encoder | Decoder | | | Ao | Ga | LK | RK | Li | Pa | Sp | St |
| ResNet-50 | dCUP | 74.45 | 32.50 | 86.75 | 60.53 | 77.78 | 71.25 | 93.25 | 51.07 | 82.93 | 72.02 |
| ConvNeXt | dCUP | 77.11 | 20.94 | 86.41 | 61.31 | 79.56 | 74.94 | 93.00 | 60.13 | 83.78 | 77.76 |
| ConvFormer | dCUP | 79.18 | 25.55 | 86.94 | 64.19 | 81.19 | 76.32 | 93.88 | 62.15 | 89.39 | 79.34 |
| 3SE+1Trans | dCUP | 82.20 | 23.98 | 84.77 | 67.67 | 89.69 | 86.06 | 94.46 | 63.24 | 87.10 | 84.62 |
| 2SE+2Trans | dCUP | 83.69 | 17.14 | 87.02 | 65.77 | 91.17 | 87.58 | **94.66** | **66.16** | 90.67 | **86.45** |
| SEFormer | None | 68.18 | 37.49 | 77.21 | 48.46 | 72.82 | 69.06 | 91.96 | 42.90 | 80.20 | 62.85 |
| SEFormer | CUP | 82.13 | 23.34 | **87.35** | 66.25 | 90.21 | 88.80 | 94.02 | 60.03 | 88.54 | 81.86 |
| Ours | | **83.90** | **15.64** | 86.11 | **69.38** | **91.66** | **89.03** | 94.38 | 64.45 | **91.12** | 85.08 |

**Notes.**
The best results are shown in bold.

### Architecture of the encoder

To validate the effectiveness of the improved encoder architecture, we conducted experiments using classical models: ResNet-50 (*He et al., 2016*), high-performance CNN network ConvNeXt (*Liu et al., 2022*), and the state-of-the-art MetaFormer (instantiated as ConvFormer model; *Yu et al., 2023*) as encoder structures. These encoders were combined with our decoder, and experiments were performed on the BCV dataset.

Results from Table 4 indicate that our SEF-UNet outperforms all three comparison models in the segmentation of all organs. The average dice score improved by 4.72%–9.45%, and the average Hausdorff distance decreased by 5.3%–16.86%, demonstrating a significant enhancement in segmentation performance.

Additionally, we explored replacing the last one or two stages of the four SEFormer blocks with Transformer blocks. The results showed that the model using SEFormer blocks in all four stages performed the best. This demonstrates that our SEF-UNet, incorporating the SEFormer model with the squeeze-and-excitation module as the encoder, has a stronger capability to learn effective representations for medical image segmentation tasks. This experiment confirms the superiority of the SEF-UNet model in terms of encoder architecture improvements.

In Tables 5 and 6, the DSC and HD for the ablation experiments on nine test cases are listed. Figures 10 and 11 respectively present the DSC and HD for different encoder and decoder experiments.

Compared to several models such as ResNet, ConvNeXt, and ConvFormer serving as encoders, our experimental model demonstrates significant improvements in segmentation performance both overall and on individual cases. When investigating the impact of the number of SEFormer modules on segmentation results, it is evident that our model, consisting of four SEFormer modules, exhibits more stable DSC and HD across all cases, achieving the overall best performance.

### Architecture of the decoder

To verify the effectiveness of the decoder architecture used in this experiment, we compared two different models. The combination of SEFormer with None Decoder represents the use

**Table 5** Average dice score (%) and average Hausdorff distance (mm) for different encoder and decoder experiments (sample 1–5).

| Framework | | Sample1 | | Sample2 | | Sample3 | | Sample4 | | Sample5 | |
|---|---|---|---|---|---|---|---|---|---|---|---|
| Encoder | Decoder | DSC | HD | DSC | HD | DSC | HD | DSC | HD | DSC | HD |
| ResNet-50 | dCUP | 74.07 | 39.51 | 80.43 | 22.44 | 72.26 | 31.20 | 62.99 | 47.58 | 70.00 | 44.50 |
| ConvNeXt | dCUP | 71.41 | 28.22 | 82.39 | 15.12 | 87.71 | 9.58 | 72.48 | 30.75 | 68.47 | 28.80 |
| ConvFormer | dCUP | 74.55 | 39.97 | 83.96 | 21.88 | 83.47 | 21.02 | 75.73 | 27.12 | 69.82 | 50.27 |
| 3SE+1Trans | dCUP | 74.63 | 43.00 | 85.39 | 17.84 | 87.68 | 32.52 | 77.09 | 28.55 | 81.70 | 25.48 |
| 2SE+2Trans | dCUP | 77.86 | 35.53 | 85.44 | **14.67** | 87.53 | 12.21 | 83.21 | 20.24 | 82.13 | 25.15 |
| SEFormer | None | 72.84 | 41.02 | 77.39 | 26.69 | 70.22 | 30.73 | 57.13 | 55.17 | 46.03 | 73.24 |
| SEFormer | CUP | 76.13 | 35.16 | 85.23 | 21.61 | 87.51 | 9.98 | 74.32 | 23.56 | 79.66 | 31.49 |
| Ours | | **78.70** | **27.97** | **85.58** | 23.04 | **87.73** | **8.99** | **84.16** | **20.18** | **82.40** | **24.31** |

**Notes.**
The best results are shown in bold.

**Table 6** Average dice score (%) and average Hausdorff distance (mm) for different encoder and decoder experiments (sample 6–9).

| Framework | | Sample6 | | Sample7 | | Sample8 | | Sample9 | |
|---|---|---|---|---|---|---|---|---|---|
| Encoder | Decoder | DSC | HD | DSC | HD | DSC | HD | DSC | HD |
| ResNet-50 | dCUP | 81.01 | 18.93 | 82.15 | 15.18 | 75.68 | 43.44 | 71.47 | 29.75 |
| ConvNeXt | dCUP | 73.92 | 21.70 | 82.53 | 17.54 | 81.05 | 15.19 | 74.06 | 21.60 |
| ConvFormer | dCUP | 82.70 | 11.48 | 84.00 | 14.74 | 81.55 | 19.65 | 76.81 | 23.83 |
| 3SE+1Trans | dCUP | 87.43 | 23.89 | 87.30 | 10.08 | 81.93 | 21.84 | 76.65 | 12.64 |
| 2SE+2Trans | dCUP | 87.43 | 7.44 | **88.42** | 3.12 | 83.28 | 18.38 | 77.92 | 17.53 |
| SEFormer | None | 80.19 | 17.86 | 64.26 | 28.78 | 73.95 | 29.27 | 71.62 | 34.68 |
| SEFormer | CUP | 87.34 | 7.24 | 88.25 | 3.23 | 82.34 | 15.93 | 78.42 | 61.85 |
| Ours | | **87.49** | **6.33** | 86.87 | **3.08** | **83.42** | **14.84** | **78.76** | **12.05** |

**Notes.**
The best results are shown in bold.

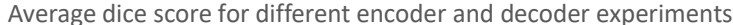

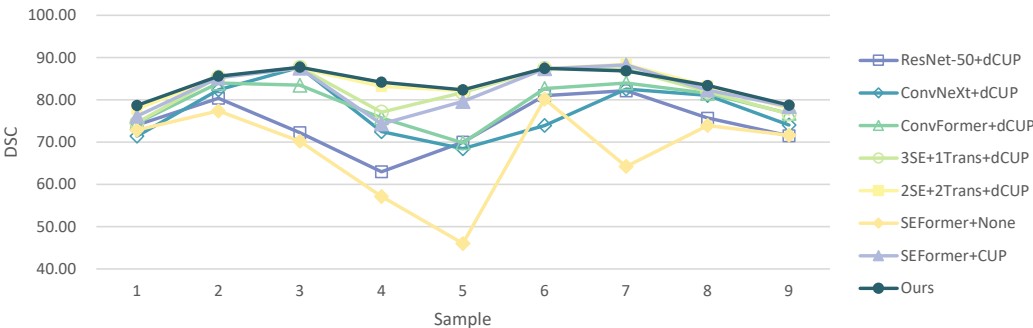

**Figure 10** Average dice score for different encoder and decoder experiments.

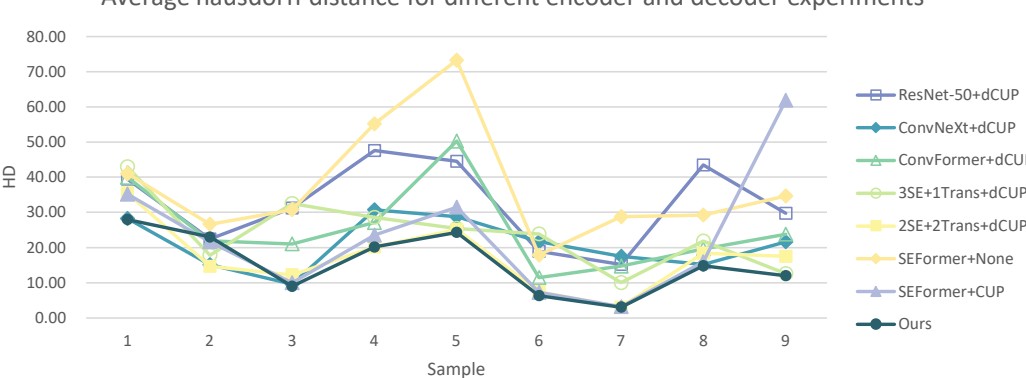

**Figure 11** **Average Hausdorff distance for different encoder and decoder experiments.**

of naive upsampling, where deep features are only upsampled to the same size as the input image without any further operations. The combination of SEFormer with CUP represents the use of an upsampling structure without the introduction of depthwise convolutional layers. After fusing the upsampled data with features passed from the encoder layers, only one convolutional layer is used to adjust the channel size of the concatenated data to a predetermined size, facilitating subsequent upsampling and feature fusion operations.

From Table 4, it can be observed that our model outperforms the naive upsampling method, with a noticeable improvement in average DSC by 15.72%. Compared to the version without the depthwise convolutional layer (CUP), the average DSC increased by 1.77%, and the segmentation results for most organs were improved.

From the results presented in Tables 5 and 6, it is evident that models without a decoder yield unsatisfactory segmentation results across all test cases. Moreover, models employing CUP as the decoder without the inclusion of deep convolutional layers demonstrate inferior overall segmentation stability compared to the combination of SEFormer and dCUP utilized in our experiment.

### Number of blocks

Our final model adopts a four-stage SEFormer architecture. In such a four-stage architecture, the first stage is mainly responsible for extracting low-level features, typically associated with the original information of the input data, such as edges, textures, *etc.*, in the image. The second stage builds upon the first stage to further extract mid-level features, such as more complex structures like the shape of objects. The third and fourth stages can extract higher-level abstract features, including object categories, scene semantics, and other advanced concepts.

To validate the effectiveness of the four-stage architecture, experiments were conducted using three-stage and two-stage architectures, as shown in Fig. 12. The model with four blocks significantly outperformed the other two models, achieving a difference of 11.63% in overall DSC. There were also noticeable differences in the segmentation results of various specific organs. Upon observation, it was found that the segmentation results for organs

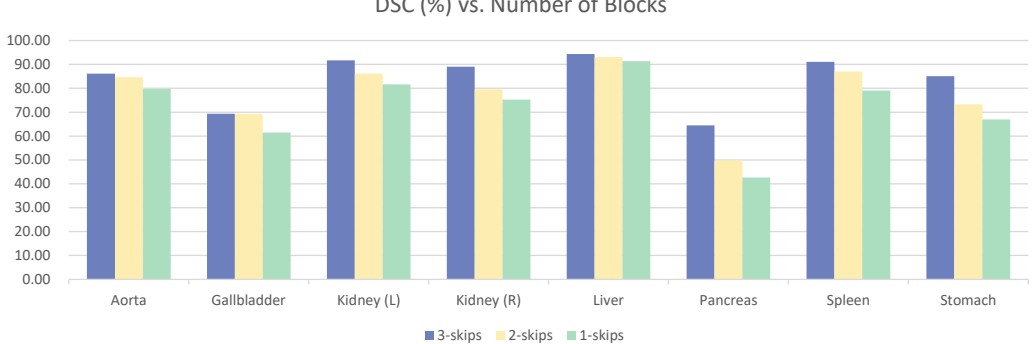

**Figure 12  Ablation experiments on the number of model stages.**

**Table 7  Average dice score (%) of different input resolutions on model results.**

| Resolution | DSC | Organs | | | | | | | |
|---|---|---|---|---|---|---|---|---|---|
| | | Ao | Ga | LK | RK | Li | Pa | Sp | St |
| 224 | 83.90 | 86.11 | 69.38 | 91.66 | 89.03 | 94.38 | 64.45 | 91.12 | 85.08 |
| 384 | **86.00** | **89.57** | **70.73** | **92.84** | **92.09** | **95.23** | **69.08** | **92.34** | **86.12** |

**Notes.**
The best results are shown in bold.

like Aorta and Liver had smaller differences, and satisfactory results could be obtained with smaller models. However, organs like Gallbladder, Kidney, Pancreas, Stomach, Spleen, *etc.*, with unclear boundaries and prone to confusion with other abdominal contents, required more complex models for clear segmentation. This further validates the effectiveness of our four-stage architecture.

### Input image size

To conserve computational resources, the model in this experiment used an input image size of $224 \times 224$, while the original dataset images were of size $512 \times 512$. Using a smaller input size theoretically may impact the model's accuracy. Therefore, we trained the model at a larger resolution of $384 \times 384$, only changing the input size, while keeping other experimental settings consistent. The results are presented in Table 7.

It can be observed that increasing the resolution from $224 \times 224$ to $384 \times 384$ raised the average dice score from 83.90% to 86.00%, showing an improvement of 2.1%. There was a noticeable enhancement in segmentation accuracy for various organs. However, this improvement comes with a significant increase in computational cost, with training time increasing by over five times. Therefore, in subsequent experiments, we consistently used a resolution of $224 \times 224$ as the input to validate the effectiveness of our experimental model.

At the same time, we conducted experiments on other comparative models using $384 \times 384$ resolution images. The results are shown in Table 8. As can be seen, all methods show a certain degree of improvement in average DSC when using $384 \times 384$ resolution input. Specifically, for smaller organs, such as the pancreas, there is a noticeable improvement. The increase for UNet++ is the smallest at 3.59%, while TransUNet shows

**Table 8  Results obtained using 384 × 384 input images across all comparative models.**

| Method | DSC | Organs | | | | | | | |
|--------|-----|--------|--------|--------|--------|--------|--------|--------|--------|
| | | Ao | Ga | LK | RK | Li | Pa | Sp | St |
| UNet | 77.69 | 88.57 | 63.34 | 82.55 | 77.26 | 94.74 | 60.15 | 85.57 | 69.29 |
| UNet++ | 79.62 | 89.27 | 70.99 | 82.66 | 75.77 | 93.56 | 60.99 | 89.10 | 74.62 |
| UNet3+ | 79.14 | 89.02 | 70.87 | 83.16 | 79.80 | 93.54 | 62.51 | 84.50 | 69.73 |
| SwinUNet | 82.87 | 88.81 | **71.94** | 86.33 | 84.53 | **96.61** | 65.00 | 90.94 | 78.80 |
| TransUNet | 84.03 | **89.78** | 71.28 | 88.19 | 85.88 | 94.59 | **71.23** | 87.92 | 83.37 |
| Ours | 86.00 | 89.57 | 70.73 | **92.84** | **92.09** | 95.23 | 69.08 | **92.34** | **86.12** |

**Notes.**
The best results are shown in bold.

the largest improvement at 13.46%. Similarly, for small-sized organs like the gallbladder and aorta, the DSC also shows an upward trend when using high-resolution input, proving that input image resolution has a significant impact on the segmentation of small-sized organs. However, for other organs, the DSC did not universally improve with increased resolution, and in some cases, it even decreased, indicating that high-resolution input does not necessarily have a positive impact on the segmentation of all organs.

*Preprocessing*

We have added a comparative experiment between pre-processed and non-pre-processed data. We will train our model using data with random flip and random rotation operations, as well as data without these operations, and then make predictions on the same test set. We have selected some slices from the prediction results of both cases for visual comparison, as shown in Fig. 13. It can be seen that, in terms of segmentation accuracy and detail precision, the results obtained by training with pre-processed data are significantly better.

## Qualitative visualization

Next, we conducted a qualitative comparative analysis of the experimental results, as shown in Fig. 14. The observations include:

(1) Pure CNN methods like UNet and ResNet-50 tend to exhibit over-segmentation or under-segmentation. For instance, in the third row, UNet produces multiple segments outside the boundary between the lungs and stomach, and ResNet-50 in the second row fails to segment a significant portion of the stomach. In comparison, our SEF-UNet method demonstrates stronger segmentation capabilities, both globally and locally.

(2) SEF-UNet achieves more precise contour segmentation. In the third row, SEF-UNet nearly approaches the ground truth for the outer edge of the lungs, while the other three models extend the outer edge into some actual organ regions, leading to increased false-positive areas. This indicates that SEF-UNet has an advantage in handling organ edge segmentation issues.

Overall, our SEF-UNet model outperforms other models in precise edge segmentation, noise suppression, and accurate shape recognition.

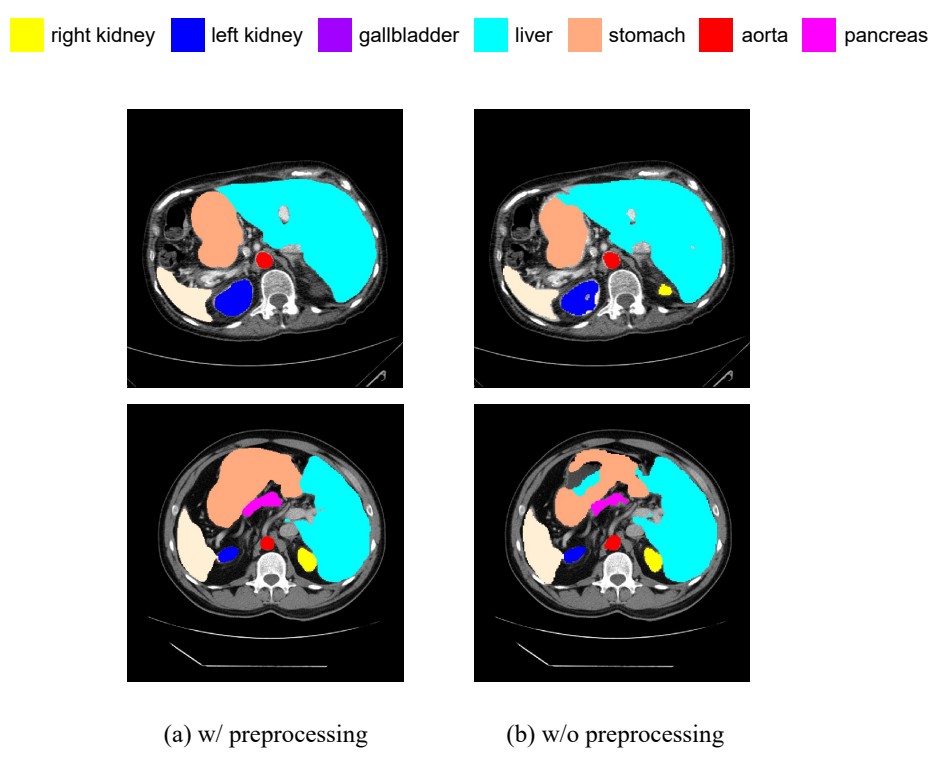

| | | | | | | | |
|---|---|---|---|---|---|---|---|
| spleen | right kidney | left kidney | gallbladder | liver | stomach | aorta | pancreas |

(a) w/ preprocessing          (b) w/o preprocessing

**Figure 13** **(A–B) Comparison of segmentation results with and without data preprocessing.**

## CONCLUSION

To address the limitations in edge segmentation accuracy, parameter count, and computational load for multi-organ abdominal segmentation, we propose the SEF-UNet model in this article. The model adopts a four-stage U-shaped architecture, with its encoder consisting of four SEFormers. Each SEFormer combines the Squeeze and Excitation module with depthwise separable convolution to reduce parameter count while capturing crucial channel information. This allows the network to focus more on important features, thereby improving model performance. In the decoder, multiple cascaded upsampling modules with depthwise convolution (dCUP) are employed to stack, significantly enhancing the ability to restore details and semantic information of the original input image with a minimal increase in computational load. Our SEF-UNet model not only improves overall segmentation performance but also excels in edge segmentation accuracy compared to other models. Compared to CNN methods, our approach achieves an improvement of 6.03% to 8.29% in average dice score, and compared to Transformer methods, an improvement of 2.59% to 3.41%. Additionally, we conducted an ablation study on the model's encoder, decoder, stage count, and input image resolution, confirming the effectiveness of our experimental model. In visualized quantitative analyses, the SEF-UNet model demonstrates more precise segmentation edges and overall superior performance compared to other models. These results indicate that our method successfully addresses the weaknesses of

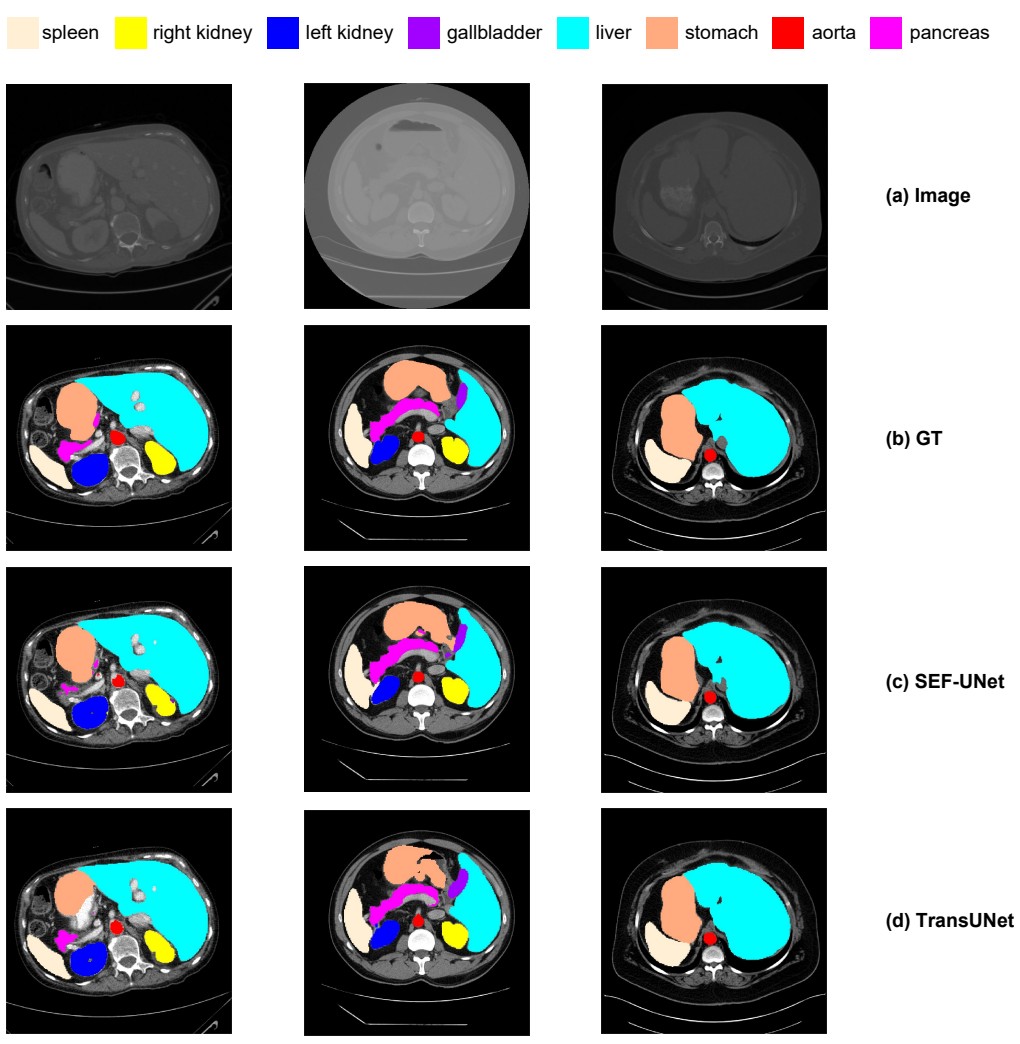

**Figure 14** **Comparison of different methods.** (A) Original image, (B) ground truth, (C) our method SEF-UNet, (D) TransUNet, (E) UNet, (F) ResNet-50 as encoder and dCUP as decoder.

previous approaches, making it more suitable for multi-organ abdominal segmentation tasks and holding promise for better application in real-world medical scenarios.

## Funding

This work was supported by the Suzhou Key Supporting Subjects, Health Informatics (No. SZFCXK202147), the Changshu Science and Technology Program (No. CS202015, CS202246), and the Changshu Key Laboratory of Medical Artificial Intelligence and Big Data (No. CYZ202301, CS202314). There was no additional external funding received for this study. The funders had no role in study design, data collection and analysis, decision to publish, or preparation of the manuscript.

## Grant Disclosures

The following grant information was disclosed by the authors:

Suzhou Key Supporting Subjects, Health Informatics: No. SZFCXK202147.

Changshu Science and Technology Program: No. CS202015, CS202246.

Changshu Key Laboratory of Medical Artificial Intelligence and Big Data: No. CYZ202301, CS202314.

## Competing Interests

The authors declare there are no competing interests.

## Author Contributions

- Yaping Zhao conceived and designed the experiments, performed the experiments, analyzed the data, performed the computation work, prepared figures and/or tables, authored or reviewed drafts of the article, and approved the final draft.
- Yizhang Jiang conceived and designed the experiments, authored or reviewed drafts of the article, and approved the final draft.
- Lijun Huang analyzed the data, authored or reviewed drafts of the article, and approved the final draft.
- Kaijian Xia conceived and designed the experiments, authored or reviewed drafts of the article, and approved the final draft.

## Data Availability

The code is available in the Supplemental File.

The code and data are available at Multi-Atlas Labeling Beyond the Cranial Vault - Workshop and Challenge, https://doi.org/10.7303/syn3193805.

Our code and model are available at figshare: Zhao, Yaping (2024). SEF-UNet.zip. figshare. Software. https://doi.org/10.6084/m9.figshare.26022625.v1.

## Supplemental Information

Supplemental information for this article can be found online at http://dx.doi.org/10.7717/peerj-cs.2238#supplemental-information.

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
