# Peer review of "SEF-UNet: advancing abdominal multi-organ segmentation with SEFormer and depthwise cascaded upsampling"

_PeerJ Computer Science, doi:10.7717/peerj-cs.2238_

## Round 0.1 · original submission · Major Revisions

While the revisions requested by the reviewers are not substantial, the work requires some integrations, particularly some extensions in terms of explaining the contribution and additional experiments.

Reviewer 1 ·

Basic reporting

Overall, a solid and novel work in the field of abdominal multi-organ segmentation. However, some extra minor experiments are expected.

Experimental design

1. “The anatomical structure of abdominal organs is relatively complex, with issues such as mutual occlusion and unclear boundaries between organs. Existing methods have room for improvement in the accuracy of organ segmentation edges.” -As one of the emphasized challenges, are there any special components in the proposed model designed to solve this problem?

2. The authors mentioned the demand of real-time and low latency. So, does the average processing time meet the standard of real-time application?

Validity of the findings

Very solid comparison results.

Notably, the authors motioned that they resized the input image from to 224x224, which is considerable for reducing the training cost. However, such preprocessing could potentially cast significant negative influence on the segmentation performance regarding some anatomical structures with relatively small size (for example, aortas and pancreas in Fig. 9 ). This phenomenon can be drawn from the ablation study conducted as well (Table.6), where there is a more evident improvement on the DSC of aortas and pancreas after a higher resolution input is feed.

Also, given that different neural network architectures may vary in their sensitivity to input resolution, would it be possible for the authors to conduct additional experiments to ascertain how changes in resolution affect the performance of competing models and on organs of different size? These findings could provide deeper insights into the optimal balance between computational efficiency and segmentation accuracy.

Additional comments

Citation style needs to be regularized.

Reviewer 2 ·

Basic reporting

The paper introduces SEF-UNet, a U-Net structured network incorporating SEFormer and Depthwise Cascaded Upsampling for Abdominal Multi-Organ Segmentation. The architecture consists of an encoder and a decoder. The encoder utilizes SEFormer Blocks, comprising SE modules (Squeeze-and-Excitation modules) and depthwise separable convolution within a MetaFormer architecture. Meanwhile, the decoder, known as dCUP, employs depthwise convolution to reduce parameters and enhance performance. Through these modules, SEF-UNet demonstrates improvements on specific benchmarks. Ablation studies are conducted to validate the effectiveness of each module individually.

Experimental design

The paper provides clear and well-defined input and output specifications. Results demonstrate improvements over prior works, and comprehensive training details are provided to facilitate reproducibility.

Validity of the findings

The main drawback of this paper is lack of novelty. All modules used to construct the proposed SEF-UNet are already well-exploited in either computer vision or medical imaging domains. As a result, it is foreseeable that by stacking these modules together, the performance can be boost compared to the baseline.

Besides, the authors try to claim things like real-time and low latency. Though the proposed model might be, providing only the number of parameters can not prove this point. A comparison on GFlops or latency is needed to support the claim.

Reviewer 3 ·

Basic reporting

All comments have been added in detail to the 4th section called additional comments.

Experimental design

All comments have been added in detail to the 4th section called additional comments.

Validity of the findings

All comments have been added in detail to the 4th section called additional comments.

Additional comments

Review Report for PeerJ Computer Science
(SEF-UNet: Advancing abdominal multi-organ segmentation with SEFormer and depthwise cascaded upsampling)

1. Within the scope of the study, multi-organ segmentation operations were performed with the recommended deep learning model based on Unet, using abdominal computed tomography images.

2. In the introduction section, the place of abdominal multi-organ segmentation in the literature and the main contributions of the study are clearly stated.

3. When the SEF-UNet architecture, consisting of SEFormer and dCUP Blocks, is examined and compared with the literature, it is observed that it has a unique structure and has the potential to make a significant contribution to the literature.

4. It was stated that Multi-Atlas Labeling Beyond the Cranial Vault (BCV) was used as the dataset in the study. It was stated that only a part of this open source dataset (only 8 out of 13) was used instead of the whole (all categories). The reason for limiting the dataset should be explained in detail. It was also stated that various augmentation procedures were performed during the training phase. It was stated that the distribution of the dataset was done randomly. It is recommended to add a table containing detailed information such as the initial data amount, training and testing data amount, post-augmentation data amount, initial and next image size for each category in the dataset.

5. Since the results obtained in segmentation problems are very dependent on the dataset, how the dataset is distributed is of great importance. It was stated in the study that the distribution was done randomly. It should be explained more clearly why cross-validation is not preferred and on what basis testing and training data are selected.

6. In the Implementation section, parameter values such as learning rate, momentum, and epoch are given for model training. Explain how these parameters were determined and whether different experiments were made.

7. The results of the proposed model are compared with other important unet-based models in the literature. There are deficiencies in the evaluation metrics obtained for the analysis of segmentation results. Please try to complete the missing metrics by reviewing the metrics obtained in similar medical segmentation studies in the literature.

8. To observe the positive effect of preprocessing/augmentation processes on the results, compare and interpret the segmentation results with the data set with/without preprocessing and/or with/without augmentation.

As a result, although the study is important in terms of the proposed original model and the problem addressed, attention should be paid to the parts mentioned above.

---

## Round 0.2 · accepted · Accept

The paper has undergone peer review and the reviewers were impressed with your work and found it to be original, well-researched, and significant to the field. Therefore, the paper has been deemed ready for publication.

Reviewer 1 ·

Basic reporting

no comment

Experimental design

no comment

Validity of the findings

no comment

Reviewer 3 ·

Basic reporting

All comments have been added in detail to the last section.

Experimental design

All comments have been added in detail to the last section.

Validity of the findings

All comments have been added in detail to the last section.

Additional comments

Review Report for PeerJ Computer Science
(SEF-UNet: Advancing abdominal multi-organ segmentation with SEFormer and depthwise cascaded upsampling)

Thank you for the revision. The final version of the paper after revision and the responses to the reviewer comments have been examined in detail. Although some of the responses are limited, I recommend that this research paper be accepted as it is since its place in the literature and originality are at an important level. I wish the authors success in their future projects. Kind regards.